# Hyaluronic Acid Oligosaccharide Derivatives Alleviate Lipopolysaccharide-Induced Inflammation in ATDC5 Cells by Multiple Mechanisms

**DOI:** 10.3390/molecules27175619

**Published:** 2022-08-31

**Authors:** Hesuyuan Huang, Xuyang Ding, Dan Xing, Jianjing Lin, Zhongtang Li, Jianhao Lin

**Affiliations:** 1Arthritis Clinic & Research Center, Peking University People’s Hospital, Peking University, Beijing 100044, China; 2Arthritis Institute, Peking University, Beijing 100044, China; 3State Key Laboratory of Natural and Biomimetic Drugs, School of Pharmaceutical Sciences, Peking University, Beijing 100191, China

**Keywords:** osteoarthritis, hyaluronic acid, hyaluronic acid oligosaccharide derivatives, ATDC5, lipopolysaccharide

## Abstract

High molecular weight hyaluronic acids (HMW-HAs) have been used for the palliative treatment of osteoarthritis (OA) for decades, but the pharmacological activity of HA fragments has not been fully explored due to the limited availability of structurally defined HA fragments. In this study, we synthesized a series glycosides of oligosaccharides of HA (o-HAs), hereinafter collectively referred to as o-HA derivatives. Their effects on OA progression were examined in a chondrocyte inflammatory model established by the lipopolysaccharide (LPS)-challenged ATDC5 cells. Cell Counting Kit-8 (CCK-8) assays and reverse transcription-quantitative polymerase chain reaction (RT-qPCR) showed that o-HA derivatives (≤100 μg/mL) exhibited no cytotoxicity and pro-inflammatory effects. We found that the o-HA and o-HA derivatives alleviated LPS-induced inflammation, apoptosis, autophagy and proliferation-inhibition of ATDC5 cells, similar to the activities of HMW-HAs. Moreover, Western blot analysis showed that different HA derivatives selectively reversed the effects of LPS on the expression of extracellular matrix (ECM)-related proteins (MMP13, COL2A1 and Aggrecan) in ATDC5 cells. Our study suggested that o-HA derivatives may alleviate LPS-induced chondrocyte injury by reducing the inflammatory response, maintaining cell proliferation, inhibiting apoptosis and autophagy, and decreasing ECM degradation, supporting a potential oligosaccharides-mediated therapy for OA.

## 1. Introduction

OA is a chronic disorder characterized by articular cartilage degradation, periarticular bone sclerosis, cysts, and inflammation. It is a major cause of inactivity, pain, and disability in middle-aged and elderly populations globally [1,2,3]. The disease affects the whole articular organ and seriously increases the public health burden, and has no effective treatment except for terminal patients to undergo joint replacement surgery [4,5,6]. OA also can affect multiple tissues surrounding the joints including articular cartilage, and chondrocytes are the major cellular component in the cartilage tissues [7,8]. The shift of chondrocyte phenotypes has been well documented to be involved in the onset and development of OA [8,9].

Lipopolysaccharide (LPS), a crucial metabolite and biomarker of gut microbiota, has been identified to cause the inflammation of OA [10,11]. ATDC5, a mouse teratocarcinoma-derived chondrogenic cell line, is regarded as an excellent in vitro cell model to explore the molecular mechanisms underlying chondrocyte biology, chondrogenesis, and skeletal development because it can recapitulate the main aspects of chondrocyte differentiation, cartilage extracellular matrix (ECM)-processing machinery, and synthesis of cartilage matrix [12,13,14]. LPS-treated ATDC5 cells have been used as an in vitro model to investigate the molecular basis of OA-related cartilage damage [15,16].

HA is a long-chain polysaccharide found in connective tissue, skin, eyes, cartilage, bone and synovial fluid. It consists of repeated disaccharide units [17,18]. In the joint, HA mainly concentrates on the surface of articular cartilage and synovial superficial layer, forming a semi-permeable barrier cartilage between synovial fluid and the synovial membrane. HA depletion is one of the pathologic factors of OA [19,20]. Moreover, the decrease of HA concentration and molecular weight (MW) in the joints is related to OA onset and progression [21,22]. Certain exogenous HA-based formulations have been demonstrated to be effective in the amelioration of OA-related symptoms (e.g., pain) by several mechanisms including the increase of HA content, inhibition of pro-inflammatory responses, and restoration of viscoelasticity of intra-articular fluid [22,23,24]. The molecular weight of HA ranges from 10^3^ to 10^7^ Da, and the complete hyaluronic acid chain can contain up to 25,000 disaccharide repeat units. Based on molecular weight, HA can be divided into several groups: HA oligosaccharide (o-HA), low molecular weight HA (LMW-HA), medium molecular weight HA (MMW-HA), high molecular weight HA (HMW-HA) and very high molecular weight hyaluronic acid (vHMW-HA) [25].

The size of HA decreases under pathological conditions, which increases the mechanical load of cartilage and induces cartilage damage [26,27]. In previous studies, HA was found to reduce the rate of chondrocyte apoptosis in OA patients [28]. In addition, it may promote not only chondrocyte proliferation and metabolism, but also promote extracellular matrix formation [19,20,29]. However, the underlying mechanisms are not well understood. Similarly, the effect of changes in the molecular mass of HA in the extracellular environment on the inflammatory processes is not fully understood [30]. Previous studies suggest that HMW-HA preparations are generally considered to have more positive pharmacological effects and stronger effects than LMW-HA or MMW-HA [27,31]. HMW-HA is considered as a physiological protective agent of cells and can act as a scaffold to assemble a proteoglycan matrix. It binds and aggregates its cell-surface receptors [32,33,34]. HA fragments (MW ranging from LMW-HA to o-HA) produced by hyaluronidases are formed in the inflammatory microenvironment and have been reported to induce defensive or pro-inflammatory responses in many cell types [35,36,37]. Endogenous HA oligosaccharides are recognized as the inflammatory response inducer by interacting with CD44 and toll-like receptor 4 (TLR-4) and 2 (TLR-2) in different types of cells [38,39,40]. However, several studies published in recent years have reported that HA fragments (including HA oligosaccharides) may fail to induce proinflammatory factors or other signaling effects in some cell types, especially in the area of OA [30,41,42].

The effects of o-HAs on the inflammation of articular chondrocytes are still unclear. Despite advances in the chemical synthesis of o-HAs [43,44,45], obtaining structurally defined derivatives is difficult. We developed a semi-synthesis route starting from acidolysis and enzymolysis of HAs to obtain HA oligosaccharides, which were further modified at the terminal position to to obtain methylation and azidation oligosaccharides. And their anti-inflammatory effects were examined in ATDC5 cells. Our results show that o-HA derivatives alleviated LPS-induced inflammation by multiple potential mechanisms.

## 2. Results

### 2.1. Study Design

Figure 1 briefly introduces the synthetic routes and molecular biological function detection of six glycosides of HA oligosaccharides.

### 2.2. Synthesis of o-HA Derivatives

#### 2.2.1. Synthesis of o-HA_2_OMe (**6**) and o-HA_2_N_3_ (**7**)

The HA disaccharides modified by β-methyl and β-azide were synthesized (Figure 2) using HA polysaccharide (MW. ~500 kDa) as the starting material. Disaccharide **1** was obtained by acid hydrolysis [46]. Acetylation in pyridine (Pyr)/acetic anhydride (Ac_2_O) resulted in medium to low yield due to the presence of a large amount of inorganic salts. A more efficient procedure for acetylation of **1** using Ac_2_O/dichloromethane (DCM) as the solvent and dimethyl aminopyridine (DMAP) as the catalyst was used to obtain **2** in 31% yield with α/β = 2.5/1 [47]. The oxazoline intermediate **3** was generated from **2** in 82% yield [48], and the disaccharides **4** and **5** were obtained through a copper(II)-mediated ring-opening reaction from the intermediate **3** in 85% and 83% yield [49], respectively. The disaccharides **6** and **7** were provided by deprotection of the disaccharides **4** and **5** through a standard procedure in 86% and 85% yield, respectively [50] (Figure 2). 

#### 2.2.2. Synthesis of o-HA_4_N_3_ (**11**) and o-HA_6_N_3_ (**12**)

As shown in Figure 2, HA polysaccharide (MW. ~500 kDa) was digested with bovine testis (or testicular) hyaluronidase (BTH) (200 mg for 10 g HA polysaccharide) in NaCl/NaOAc buffer (pH = 5.0) for 7 days to generate HA tetrasaccharide and hexasaccharide mixture [51]. The mixture was converted into β-azide derivatives through a (2-chloro-N,N′-1, 3-dimethylimidazolium chloride)-mediated nucleophilic reaction with NaN_3_ in water without protecting groups [52]. Through a continuous esterification method by sequentially methylating and acetylating the degradation mixture, the tetrasaccharide **8** and hexasaccharide **9** were isolated via chromatography in a three-step overall yield of 42% and 34%, respectively. The tetrasaccharide **10** and hexasaccharide **11** were provided by the deprotection of tetrasaccharide **8** and hexasaccharide **9** through a standard procedure in 85% and 82% yield, respectively (Figure 3). 

#### 2.2.3. Synthesis of o-HA_4_OMe (**14**) and o-HA_6_OMe (**15**)

As shown in Figure 4, the HA tetrasaccharide and hexasaccharide mixture digested by BTH was dissolved in a high concentration of HCl/MeOH (0.5 M) solution to convert into β-methyl derivatives, following a standard procedure of acetylation. The β-methyl derivatives **12** and **13** were separated in a three-step overall yield of 21% and 19%, respectively (Figure 4). The tetrasaccharide **14** and hexasaccharide **15** were provided by the deprotection of tetrasaccharide **12** and hexasaccharide **13** in 85% and 82% yield, respectively.

### 2.3. Evaluation of the Cytotoxicity and Pro-Inflammatory Activity of o-HA Derivatives in ATDC5 Cells

Before evaluating anti-inflammatory activity, we tested the cytotoxicity of the synthetic o-HA derivatives in ATDC5 cells using the CCK-8 assay. The cell inhibitions of the six o-HA derivatives at different doses (0, 1, 10, 100 or 1000 μg/mL) were examined after 24 h. o-HA_2_N_3_ and o-HA_4_N_3_ showed little effect on cell viability (Figure 5A). o-HA_6_N_3_ and o-HA_6_OMe demonstrated good safety at concentrations of less than 1000 μg/mL. The other two derivatives o-HA_2_OMe and o-HA_6_OMe showed a dose-dependent inhibition of ATDC5 cell viability. This showed that the β-azide derivatives exhibited better safety compared to β-methyl derivatives and the safe concentrations between 0–100 μg/mL of the β-azide derivatives and o-HA_6_OMe were used in the following experiments.

In the past study [36], HA oligosaccharides (40 μg/mL) were reported to induce inflammatory response of mouse chondrocytes. We tested the related activity on ATDC5 cells after treatment with the synthetic o-HA derivatives at 40 μg/mL. After treatment with o-HA derivatives for 24 h, RT-qPCR for IL-1β (Figure 5B) and IL-6 (Figure 5C) was performed to confirm the inflammatory effect. Results showed that all six derivatives had no inflammation causing effect on the mRNA levels of these two inflammatory cytokines at 40 μg/mL. Moreover, to verify whether the o-HA derivatives exhibit inflammatory activity, we added the experiment related to THP-1 cells and obtained similar results (Appendix A).

### 2.4. Evaluation of the Protective Effect of Different HA Derivatives on LPS-Induced Inflammatory Injury of ATDC5 Cells 

To investigate the anti-inflammatory potential of the synthetic o-HA derivatives, we tested their effects on the expression of pro-inflammatory cytokines IL-1β and IL-6 in LPS challenged ATDC5 cells. As shown in Figure 6A,D, compared with the control group, the mRNA levels of IL-1β and IL-6 were significantly elevated by LPS. All the synthetic derivatives groups showed decreased mRNA levels of both pro-inflammatory markers (IL-1β and IL-6) compared with the LPS group. The results also suggested that the disaccharides and hexasaccharides exhibited higher potency against LPS-induced inflammation. Considering the lower cytotoxicity of hexasaccharide derivatives, o-HA_6_N_3_ and o-HA_6_OMe were selected for further investigation. RT-qPCR was used to investigate the anti-inflammatory effects of o-HA hexasaccharides, HMW-HAs (HMW-HA1 with MW. 800~1500 kDa and HMW-HA2 with MW. 1800 kDa), and o-HA_6_ which were obtained from the enzymolysis of HA. We found that the mRNA levels of two HMW-HA groups were significantly decreased compared with the control group, and o-HA_6_ could also reduce the levels of IL-1β and IL-6 mRNA (Figure 6B,E). Compared with o-HA_6_, the other two synthetic o-HA_6_ derivatives showed higher anti-inflammatory activity at a similar level of HMW-HA. 

To further verify the effect on inflammatory response, enzyme-linked immunosorbent assay (ELISA) was used to determine the release of pro-inflammatory cytokines (IL-1β and IL-6) (Figure 6C,F) after treatment with different HA derivatives (HMW-HA1/2, o-HA_6_, o-HA_6_N_3_ and o-HA_6_OMe). It was found that the release of inflammatory cytokines decreased significantly after 24 h treatment with different HA derivatives. For the inhibition of IL-1β, all derivatives showed high potency and had no significant difference, except that the difference between HWM-HA2 and o-HA_6_N_3_ was significant (*p* < 0.05) (Figure 6C). For the inhibition of IL-6, o-HA_6_ and o-HA_6_ derivatives showed superior activity to HMW-HA2 with statistically significant difference (*p* < 0.05, *p* < 0.01), but showed no difference among the three o-HA_6_ derivatives (Figure 6F). The anti-inflammatory activity of different HA derivatives was also confirmed by Western blot analysis (Appendix A). We also examined the anti-inflammatory activity of different HA derivatives in THP-1 cells by RT-qPCR, and the result confirmed the anti-inflammatory activities of different HA derivatives (Appendix A). However, a different activity pattern among the tested group was observed compared with that in ATDC5 cells, which may be related to the different cell line and experimental condition. Further experimental study is needed.

### 2.5. The Proliferation Effect of Different HA Derivatives on LPS Challenged ATDC5 Cells

To confirm the proliferation effects of different HA derivatives on LPS-challenged ATDC5 cells, flow cytometry was used to evaluate the 5-Ethynyl-2'-deoxyuridine (EdU)-positive cell rate. As shown in Figure 7A,B, the percentage of EdU-positive cells decreased significantly after LPS was added. In contrast, treatments with different HA derivatives (HMW-HA1, HMW-HA2, o-HA_6_, o-HA_6_N_3_ and o-HA_6_OMe) for 24 h significantly increased the percentage of EdU-positive ATDC5 cells, indicating that various HA derivatives can restore cell proliferation ability (Figure 7C–G). However, according to the statistical results of Figure 7H, there was no significant difference in the alleviating effects of different HA derivatives on LPS-induced inflammatory injury.

### 2.6. Anti-Apoptotic Effect of Different HA Derivatives on LPS Challenged ATDC5 Cells

To study the effects of different HA derivatives on apoptosis, the TUNEL (TdT mediated dUTP Nick End Labeling) assay was used to analyze apoptosis of ATDC5 cells after LPS treatment. TUNEL assay showed that LPS-induced apoptosis of ATDC5 cells with a significantly increased apoptotic cell ratio. All HA derivatives (HMW-HA1, HMW-HA2, o-HA_6_, o-HA_6_N_3_ and o-HA_6_OMe) significantly inhibited LPS-induced apoptosis, among which HMW-HA1 showed the most potent effect. o-HA_6_N_3_ and o-HA_6_OMe showed moderate alleviating effect on apoptosis, and the difference between o-HA_6_N_3_ and o-HA_6_OMe was statistically significant (*p* < 0.05) (Figure 8A,B). The Bcl-2 gene is known as an apoptosis-suppressing gene. Bcl-2 can homodimerize and form heterodimers with Bax, in which Bcl-2 acts as the apoptosis inhibitor and Bax acts as the apoptosis promoter. Therefore, we also analyzed the protein expression of Bcl-2 and Bax by Western blot assay. The result showed that the expression of Bcl-2 was selectively increased and that of Bax was decreased, indicating increased cell resistance to apoptosis due to the protective effect of different HA derivatives. Compared with the control group, Bcl-2 expression was significantly decreased and Bax expression was significantly increased in the LPS group. After treatment with different HA derivatives in the LPS group, the expression of Bax decreased significantly. However, o-HA or o-HA derivatives did not significantly alter the expression of Bcl-2 in the LPS group, but HMW-HA treatment restored the expression of Bcl-2 (Figure 8C).

### 2.7. Anti-Autophagy Effect of Different HA Derivatives on LPS Challenged ATDC5 Cells

The most commonly used indicator for autophagy detection is the microtubule-associated protein light chain 3 (LC3) [53]. LC3 conversion (LC3-I to LC3-II) can be detected by immunofluorescence analysis because the level of LC3-II correlates with the number of autophagosomes. Our results showed that compared with control group LPS promoted the autophagy process of ATDC5 cells (Figure 9A,B). Most of the autophagy structures appeared at the outside of the cell membrane after LPS injury, showing the highest fluorescence intensity. After treatment with the HA derivatives, the aggregation of autophagy around the cell membrane was alleviated significantly with decreased fluorescence intensity. Different HA derivatives showed prominent anti-autophagy effects revealed by the decreased immunofluorescence intensity, but the differences among them were not statistically significant. The endogenous LC3 protein exhibits two bands expressed from the same mRNA, LC-I with MW of 18 kDa and LC-II with MW of 16 kDa. The ratio of LC3-II/LC3-I is closely correlated with the number of autophagosomes, serving as a reliable indicator of autophagosome formation [54]. Therebefore, we used Western blot assay to measure the LC3-II/LC3-I ratio in the study. The ratio of LC3-II/LC3-I increased significantly after LPS treatment (Figure 9C). In comparison, after treatment with 40 μg/mL of different HA derivatives for 24 h, the ratio of LC3-II/LC3-I decreased significantly. The results of o-HA and o-HA derivatives suggest that they may be more beneficial in inhibiting autophagy in LPS-stimulated ATDC5 cells. In addition, the autophagy-inhibiting effect of HA derivatives was investigated in the presence of an autophagy inhibitor (Appendix A). The results showed that in the absence of the autophagy inhibitor, o-HA derivative (o-HA_6_OMe) and HMW-HA (HMW-HA1) had similar decreasing effects on the LC3-II/LC3-I ratio increase caused by LPS (*p* < 0.0001). However, after the addition of the inhibitor, the decreasing effect on the ratio was weakened. HMW-HA1 showed a decreasing effect that was not statistically significant, and o-HA_6_OMe showed a statistically significant (*p* < 0.05) decreasing effect. 

### 2.8. Alleviation of the ECM Degradation by Different HA Derivatives

ECM degradation is an important pathogenic mechanism of OA, inducing a degenerative cycle of inflammation and degradation of cartilage by proteinases such as matrix metalloproteinases (MMPs). MMP13 plays a key role in cartilage degradation in OA, destroying not just type II collagen, but proteoglycan, type IV and type IX collagen, as well as osteonectin and basement membrane [55]. Proteoglycans (such as Aggrecan), glycoproteins (such as COMP), polysaccharides and fibrins (such as collagen type II alpha 1 (COL2A1) and elastin) synthesized by chondrocytes are the main components in ECM, and they are closely related to the progress of OA [56,57,58]. To evaluate the effects of different HA derivatives on ECM degradation we measured the levels of MMP13, COL2A1 and Aggrecan by Western blot to determine the protein indexes of cell matrix and collagen. Results of Western blot assay showed that stimulated by LPS (Figure 10A), MMP13 was significantly increased, and the different HA derivatives reduced the increased level; especially, o-HA_6_OMe showed high potency (Figure 10B). COL2A1 and Aggrecan of ATDC5 cells decreased significantly under LPS stimulation. Moreover, the treatment of different HA derivatives (HMW-HA1, HMW-HA2, o-HA_6_ and o-HA_6_N_3_, except o-HA_6_OMe group) reversed the decreasing trend of COL2A1 at different degrees. However, for Aggrecan, almost all the different HA derivatives did not show protective effects except for o-HA_6_N_3_, which showed a slightly significant protective effect (Figure 10C,D).

## 3. Discussion

OA is a chronic degenerative disease of the joints that leads to pain and loss of mobility [59]. Intra-articular injection of high molecular weight HA to replace synovial fluid that has lost its viscoelastic properties is widely used in the treatment for OA [60]. HA plays a dual role in the process of inflammation and damage of cartilage, as a pro-inflammatory molecule or an anti-inflammatory molecule [31] It basically depends on the molecular weight of HA. In addition, a study of rheumatoid arthritis have shown that HA modulates inflammation based on its molecular weight, and HMW-HA was reported to inhibit the production of pro-inflammatory mediators and down-regulate NF-kB by binding to ICAM-1 [61]. It is well known that the immune system is essential in protecting the body from environmental threats and pathogens [62]. However, excessive activation of the immune system can lead to a variety of chronic and acute inflammation and cause a variety of diseases. As members of the pattern recognition receptor (PRRs) family, 13 Toll-like receptors (TLRs) have been identified in laboratory mice and 10 functional TLRs identified in humans. These recognize various PAMPs (pathogen-associated molecular patterns), including lipopolysaccharides (LPS) [63]. LPS-induced cartilage inflammation model is often used to mimic OA in humans, which has several advantages including convenient operation and inflammatory effects consistent with the OA disease. Multiple studies have shown that LPS exposure can trigger marked up-regulation in the expression levels of pro-inflammatory cytokines (e.g., IL-1β, IL-6), pro-apoptotic protein Bax, and a noticeable reduction in the expression of anti-apoptotic protein Bcl-2 in ATDC5 cells [64]. However, although HMW-HAs had been demonstrated to be effective in reducing the inflammation induced by LPS, the effects of HA fragments such as o-HAs or their derivatives on LPS-induced cartilage inflammation are still not clear.

Great advances have been made in the chemical synthesis of HA oligosaccharides in recent years, but the extremely long routes, low yields, small scale and high cost limit the biological activity evaluations of HA oligosaccharides [43,44,45]. In addition, hyaluronic acid can be degraded into oligosaccharides by acids and enzymes, but the main limitation of inhomogeneity of degradation products remains [46,65,66]. In our study, HA oligosaccharides were prepared in only four to five steps after acidolysis and enzymolysis of the HA polysaccharide. In addition to obtaining structurally defined oligosaccharides, HA oligosaccharides were modified by β-azide and β-methyl at the terminal position. The effects of these o-HA compounds on OA development were investigated in the LPS-induced ATDC5 cell model. Our results showed that the six o-HAs or their derivatives had similar anti-inflammatory activity, but some differences remained. The following general conclusions were obtained: (1) hyaluronic acid oligosaccharides derivatives exhibited anti-inflammatory activity in a manner as effective as HMW-HAs in an LPS-challenged inflammatory model; (2) the cytotoxicity of methylation derivates seemed to be higher than that of azide derivatives; (3) the modifications at the terminal group might benefit the anti-inflammatory activity.

We found there were significant differences in cell viability at 24 h between the cells treated with o-HAs modified with β-methyl group or with β-azide group. Methylated disaccharides and tetrasaccharides showed a certain cytotoxicity, while the cytotoxicity of methylated hexasaccharides and azide-modified hexasaccharides remained low. It is generally believed that the longer the sugar chain of HA, the more stable the compound and the less toxic it is to the cells. To be consistent, we selected two hexasaccharides with terminal modifications for activity comparison with HMW-HA and unmodified o-HA_6_. 

Although different cell sources and models were used, our results are consistent with a previous report that various HA fragments with low endotoxin content (including o-HAs) cannot induce or enhance proinflammatory cytokine release from chondrocytes [42]. Another study reported that exogenous HA fragments up to about 40 kDa in molecular weight do not show pro-inflammatory activity in human articular chondrocytes [30]. Contrary to most studies in this field in recent years, this suggests that the accumulation of HA-fragment in inflammatory tissues may be the result of inflammation rather than a driver of it. This conclusion is challenged by concerns of the purity and endotoxin content of earlier samples. Our study has now confirmed that the HA oligosaccharide derivatives exhibited anti-inflammatory activity comparable to HMW-HA. A recent study showed that the enzymatically obtained HA disaccharide, ΔHA2, inhibited LPS-induced inflammation in different cell lines (including THP-1 cells). In addition, hyaluronic acid oligosaccharides (2mer–8mer) prepared by acid hydrolysis can reduce LPS-induced inflammatory damage to varying degrees in RAW 264.7 cells [67]. Our experiment verified the activity of different HA derivatives against LPS-induced inflammatory injury in THP-1 cells. However, evidence for the therapeutic application of hyaluronic acid oligosaccharides and their derivatives in the treatment of osteoarthritis is still lacking. Admittedly, our study of the ATDC5 cell line is only the first step in advancing the application of o-HAs and their derivatives in OA.

Chondrocyte apoptosis and autophagy have been implicated in the pathogenesis of OA [68,69]. Cell apoptosis is mediated by multiple pathways, including caspase family members, of which anti-apoptotic protein Bcl-2 and pro-apoptotic protein Bax regulate the release of apoptotic activators such as cytochrome C by controlling mitochondrial membrane permeability. The Bax dimer opens channels in the membrane and increases permeability. Bcl-2 and Bax form a heteromer and reduce permeability. Therefore, the increased expression of Bcl-2 and the decreased expression of Bax indicate that the cells are more resistant to apoptosis [68]. Our study showed that LPS facilitated cell apoptosis, while these effects were inhibited by different HA derivatives. The expression of Bax decreased significantly after treatment with different HAs derivatives in the LPS group. However, HMW-HAs restored the protein expression level of Bcl-2. And o-HAs or o-HAs derivatives did not change the expression. This result may be related to the different molecular mechanisms underlying the actions of oligosaccharides and HWM-HA. The TUNEL assay indicated HMW-HA had the most significant anti-apoptotic effect compared with o-HA_6_ derivatives, followed by o-HA_6_N_3_ and o-HA_6_OMe. The effect of o-HA_6_N_3_ was higher than that of o-HA_6_OMe. 

During autophagy, the target structures are sequestered by phagophores, matured into autophagosomes, and finally delivered to lysosomes for degradation. Autophagy is involved in the pathophysiology of many diseases, and the study of its regulation helps understand the outcome of many diseases. Lysosomal inhibitors that primarily block lysosomal degradation, such as Bafilomycin A1 (Baf-A1), protease inhibitors, and chloroquine (CQ) [70], are used to block the final step of the autophagy process. We found that the anti-autophagy effects of HA derivatives were weakened in the presence of the autophagy inhibitor. After administration of lysosomal inhibitor, o-HA_6_OMe showed stronger inhibition. Compared with HMW-HA1, the number of autophagosomes was significantly reduced. (Appendix A). Inflammation can induce the remodeling of ECM of cartilage, which is closely associated with OA development [71]. ECM, a complex molecular network surrounding the cells, plays vital roles in the maintenance and regulation of cell phenotypes, cell/tissue functions, and tissue homeostasis [72,73]. Type II Collagen (COL2) is secreted by chondrocytes in the form of soluble collagen. Its supramolecular structure is generally composed of three identical alpha chains (triple helices), which is held together by interchain or intra-chain hydrogen bonds [74]. Aggrecans are the most abundant proteoglycans in cartilage, which is mainly composed of anionic sulfated glycoaminoglycan (GAG), chondroitin sulfate (CS) and keratin sulfate (KS) covalently bound with the core protein backbone. Aggrecans only exist as aggregates within the ECM. For example, the components mentioned above are non-covalently linked to another long GAG chain, hyaluronic acid, to form larger protein aggregates via “link proteins” [75]. Our results of Western blot assay suggested that the effect of HA oligosaccharide derivatives on ECM supramolecular structure in OA may depend on the down-regulation of MMP, a degradation factor, and restoration of the three-dimensional structure of ECM (up-regulation of COL2A1). The abnormal remodeling of ECM is related to the pathogenesis of multiple diseases [72,76]. Moreover, the destruction and disorganization of cartilage ECM is a common feature of OA [77]. Chondrocytes play vital roles in the production, maintenance, and remodeling of cartilage ECM by inflammatory mediators, growth factors, and enzymes [78]. In OA, anomalous cartilage ECM signals can be transmitted to chondrocytes and lead to the abnormality in the phenotypes and behaviors of chondrocytes, which further triggers ECM remodeling and cartilage disturbance, and facilitates OA progression [71,77]. Moreover, previous studies have shown that HA can exert its biological and pharmacological activities by regulating the chemical and physical properties of ECM [79,80]. Given the impact of the o-HA derivates on inflammation, we further demonstrated that these o-HA derivates could exert their protective functions by altering ECM composition and structure. Consistent with a previous report [64], our study also demonstrated that the expression levels of aggrecan and COL2A1 were notably reduced, and ECM degradation-related protein MMP13 was markedly increased, in ATDC5 cells following LPS stimulation. In conclusion, the different HA derivatives selectively alleviated LPS-induced ECM destruction in ATDC5 cells by abating the expression of ECM-related proteins.

The o-HA derivatives with β-azide or β-methyl modification were synthesized with a shortened procedure; these compounds can be easily produced at a large scale. These o-HA derivatives inhibited LPS-induced pro-inflammatory responses and cell apoptosis and autophagy in ATDC5 cells. They also restored the inhibition of cell proliferation caused by LPS inflammatory injury. Moreover, these o-HA derivatives partially reversed the effects of LPS on the expression of ECM proteins (i.e., COL2A1, aggrecan) and ECM degradation-related proteins (i.e., MMP13) in ATDC5 cells. These data suggest that o-HA derivatives could alleviate LPS-induced chondrocyte injury by reducing inflammation and altering ECM components, suggesting the potential therapeutic value of these HA oligosaccharides in OA management. 

## 4. Materials and Methods

### 4.1. Material and Reagents

LPS (Escherichia coli 055: B5) was obtained from Sigma-Aldrich (St. Louis, MO, USA). HMW-HA (0.8–1.5 × 10^6^ Da, 1.8 × 10^6^ Da) was obtain from Meilunbio (Dalian, China). CCK-8 and One Step TUNEL Apoptosis Assay Kit were purchased from Beyotime (Shanghai, China). DAPI solution and FITC conjugated Goat Anti-Rabbit IgG (H + L) was obtained from Servicebio (Wuhan, China). All primary antibodies used in this study were from Proteintech Group (Chicago, IL, USA) except the anti-β-actin (Medical Discovery Leader (MDL) Biotechnology, Beijing, China), anti-IL-1β (Abcam, Cambridge, UK) and anti-COL2A1 (Santa Cruz Biotechnology, Dallas, TX, USA). Bafilomycin A1 (Baf-A1) was purchased from Selleck (Houston, TX, USA). The reagents used for synthesis of o-HA derivatives were of analytical grade.

### 4.2. Chemical Synthesis

The detailed synthesis procedures of HA oligosaccharides are shown in Figure 2, Figure 3 and Figure 4 and Appendix A. The ^1^H NMR, ^13^C NMR, HRMS and HPLC spectra of the compounds are presented in Appendix A. 

### 4.3. Cell Culture and Treatment

The mouse chondrogenic cell line ATDC5 obtained from FuHeng Cell Center (Shanghai, China). ATDC5 cell line was cultured in DMEM medium (Hyclone, Thermo Scientific, Waltham, MA, USA) supplemented with 10% fetal bovine serum (Gibco, Thermo Scientific) and penicillin/streptomycin solution (Solarbio Science & Technology, Beijing, China) at 37 °C in the humidified atmosphere of 5% CO_2_ in air. For ATDC5 inflammation damage, the medium was replaced with medium containing 10% FBS and LPS Supplement (final concentration 10 μg/mL) for inflammatory injury of chondrocytes.

Human monocyte cell line THP-1 cells were obtained from Procell Life Science & Technology (Wuhan, China). RPMI-1640 was a suitable medium for THP-1 cells and was obtained from Gibco. THP-1 cells in FBS concentration and CO_2_ incubator conditions were the same as ATDC5 cells. PMA (Phorbol 12-myristate 13-acetate, 25 ng/mL, Meilunbio, China) induced THP-1 monocytes to differentiate into macrophages. After 48 h of induction, THP-1 cells were incubated in RPMI-1640 containing LPS (final concentration 200 ng/mL) or LPS plus different HA derivates (40 μg/mL) for 24 h.

### 4.4. CCK-8 Assay

The CCK-8 assay was performed using the CCK-8 kit (Beyotime) to measure the cytotoxicity of six o-HA derivatives in ATDC5 cells. Briefly, cells were seeded into 96-well plates and then treated with different doses of HA oligosaccharides. At 24 or 72 h after treatment, 10 μL of CCK-8 solution was added to each well. After 1 h of incubation, the absorbance was measured at 450 nm. Three replicates were performed for each group. 

### 4.5. RT-qPCR Assay

RNA was extracted from ATDC5 and THP-1 cells using the Trizol reagent (Thermo Scientific) following the protocols of the manufacturer. RNA was reversely transcribed into cDNA using the SuperScript III First-Strand Synthesis SuperMix for RT-qPCR (Thermo Scientific). Real-time quantitative PCR reactions were performed on the Applied Biosystems StepOne Real-time PCR system (Thermo Scientific) using the SYBR Select Master Mix (Thermo Scientific) and corresponding quantitative PCR primers under the reaction conditions: 95 °C for 5 min and 40 cycles of 95 °C for 10 s, 58 °C for 20 s, and 72 °C for 20 s. The quantitative PCR primers for ATDC5 were as follows: 5′-CTCCTGAGCGCAAGTACTCT-3′ (forward) and 5′-TACTCCTGCTTGCTGATCCAC-3′ (reverse) for β-actin; 5′-GTGAAATGCCACCTTTTGACA-3′ (forward) and 5′-GATTTGAAGCTGGATGCTCT-3′ (reverse) for IL-1β; 5′-CTTCCATCCAGTTGCCTT-3′ (forward) and 5′-CTGTGAAGTCTCCTCTCCG-3′ (reverse) for IL-6. The quantitative PCR primers for THP-1 were as follows: 5′-TCCTCCTGAGCGCAAGTACTCC-3′ (forward) and 5′-CATACTCCTGCTTGCTGATCCAC-3′ (reverse) for β-actin; 5′-CTCTCTCCTTTCAGGGCCAA-3′ (forward) and 5′-GCGGTTGCTCATCAGAATGT-3′ (reverse) for IL-1β; 5′-ACTCACCTCTTCAGAACGAATTG-3′ (forward) and 5′-CCATCTTTGGAAGGTTCAGGTTG-3′ (reverse) for IL-6.

### 4.6. ELISA 

After treating the cells according to the experimental requirements, cell culture supernatants were collected, and the expression of IL-1β and IL-6 was detected by ELISA using Mouse IL-1β (BOSTER, Wuhan, China) and Mouse IL-6 ELISA kits (BOSTER, Wuhan, China) according to the manufacturer’s instructions.

### 4.7. Western Blot Assay 

Protein was extracted from ATDC5 cells using the Protein Extraction Solution (MDL) supplemented with protease inhibitor (MDL). A BCA protein analysis kit (MDL) was used to quantify protein concentration. Protein was separated by 10% sodium dodecyl sulfate polyacrylamide gel electrophoresis (SDS-PAGE) and transferred to polyvinylidene fluoride membranes (0.22 μm; Millipore, Bedford, USA). After blocking non-specific interactions, the membranes were incubated overnight at 4 °C with primary antibody against β-actin (MDL), Aggrecan, Bax, Bcl-2, LC3 (Proteintech), and COL2A1(Santa Cruz). Next, the membranes were incubated for 1 h with corresponding HRP*Polyclonal Goat Anti-Rabbit IgG (H + L) (MDL) conjugated with horseradish peroxidase at room temperature. Finally, the proteins were stained and captured using Pierce ECL Western Blotting Substrate (Thermo Scientific) on the Bio-Rad ChemiDoc MP imaging system (Bio-Rad Laboratories, Hercules, CA, USA). 

### 4.8. TUNEL Assay

Cells were seeded into 24-well plates containing the chamber slides and cultured overnight. Next, cells were treated with LPS (10 μg/mL) for 24 h alone or in combination with different HA derivatives (40 μg/mL) for an additional 24 h. The cell apoptotic pattern was detected using the One Step TUNEL Apoptosis Assay Kit (Beyotime) according to the protocols of the manufacturer. Briefly, the cells were fixed with 4% paraformaldehyde in phosphate buffered saline for 30 min, rinsed with PBS, and permeabilized with 0.3% TritonX-100 for 5 min on ice followed by TUNEL for 1 h at 37 °C in the dark. After rinsing f3 times with a PBS solution containing 0.1% Triton X-100 and 5 mg/mL BSA, cells were stained with DAPI solution (Servicebio) for 5 min at room temperature. The cells with green fluorescence were detected as apoptotic cells. After washing with PBS again for three times, cells were observed under a fluorescence microscope (Leica, Wetzlar, Germany). TUNEL-positive apoptotic cells were labeled with FITC (green), and cell nuclei were stained with DAPI (blue).

### 4.9. Immunofluorescence (IF) Assay

Cells were seeded into 24-well plates containing the chamber slides and cultured overnight. Cells were treated with 10 μg/mL of LPS, followed by 40 μg/mL of different HA derivatives for 24 h. The cells growing on the chamber slides (cell-climbing films) were fixed with 4% paraformaldehyde and blocked with goat serum (Beyotime) for 1 h at 37 °C. Next, cells were incubated overnight at 4 °C with anti-LC3 primary antibody (Proteintech). Subsequently, cells were incubated with fluorescence-labeled goat anti-rabbit secondary antibody (MDL) for 1 h in the dark at 37 °C. Then, cells were stained with DAPI solution (Servicebio) for 10 min in the dark. Finally, cells were imaged under the Leica DM3000 fluorescent microscope.

### 4.10. EdU Flow Cytometry Assay

Cell proliferative activity was examined using the Cell-Light EdU Apollo567 In Vitro Flow Cytometry Kit (Ribo Biotechnology, Guangzhou, China) according to the instructions of the manufacturer. Briefly, cells were treated with 10 μg/mL of LPS, then 40 μg/mL of different HA derivatives for 24 h. Next, 50 μM EdU solution was added to cells. After 2 h of incubation, cells were fixed with 4% paraformaldehyde for 20 min and treated with 2 mg/mL glycine for 5 min. The cells were then incubated with 0.5% TritonX-100 solution for 10 min and stained with Apollo staining solution for an additional 10 min in the dark at room temperature. After rinsing for 3 times with PBS, the cells were suspended again and analyzed by flow cytometry.

### 4.11. Statistical Analysis

Data were analyzed using the GraphPad Prism software Version 8.3.0 (San Diego, CA, USA). Results are shown as mean ± standard deviation. The difference among groups was analyzed using one-way or two-way ANOVA and the Turkey test. A statistically significant difference was defined at *p* < 0.05.

## Figures and Tables

**Figure 1 molecules-27-05619-f001:**
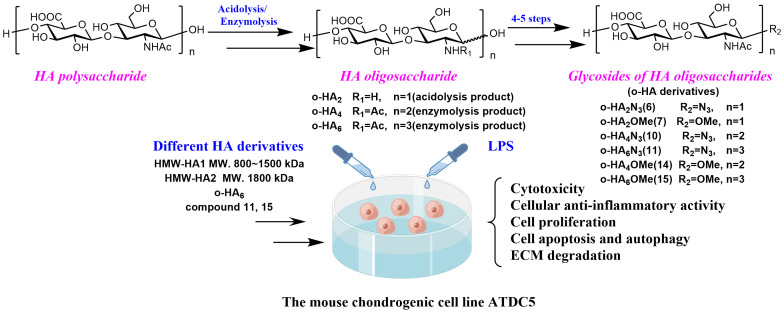
Synthesis of glycosides of HA oligosaccharides and evaluation of their biological activity in ATDC5 cells.

**Figure 2 molecules-27-05619-f002:**
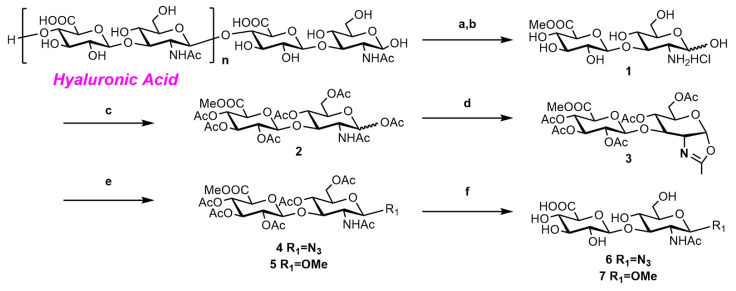
Synthesis of o-HA_2_OMe (**6**) and o-HA_2_N_3_ (**7**). Reagents and conditions: (**a**) 0.5 M HCl, 80 °C, 2 d; (**b**) 0.06 M HCl, MeOH, 4 °C, 4 d; (**c**) Ac_2_O, Et_3_N, DMAP, DCM, 2 h, α/β = 2.5/1, 31% for 3 steps; (**d**) TMSOTf, DCM, 4 °C to room temperature (rt), 82%; (**e**) CuCl_2_, MeOH/TMSN_3_, CHCl_3_, reflux, 82% for **4**, 85% for **5**; (**f**) LiOH/H_2_O_2_, THF/H_2_O, −5 °C to rt, then 4 M NaOH, MeOH, 0 °C to rt, 85% for **6**, 86% for **7**.

**Figure 3 molecules-27-05619-f003:**
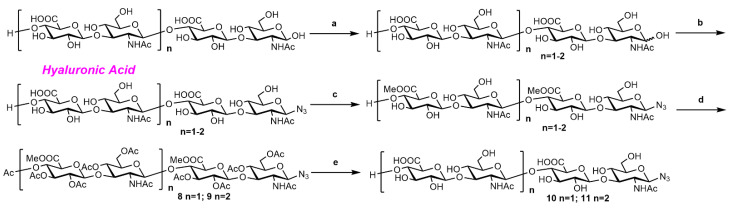
The synthesis of o-HA_4_N_3_ (**11**) and o-HA_6_N_3_ (**12**). Reagents and conditions: (**a**) 2.5% bovine testis hyaluronidase, NaOAc/NaCl buffer, pH 5.0, 37 °C, 7 days; (**b**) DMC, N-methylmorpholine, NaN_3_, H_2_O, 0 °C to rt; (**c**) 0.06 M HCl, MeOH, 4 °C, 4 d; (**d**) Ac_2_O, Et_3_N, DMAP, DCM, 2 h, 42% for **8**, 34% for **9**; (**e**) LiOH/H_2_O_2_, THF/H_2_O, −5 °C to rt, then 4 M NaOH, MeOH, 0 °C to rt, 85% for **10**, 86% for **11**.

**Figure 4 molecules-27-05619-f004:**
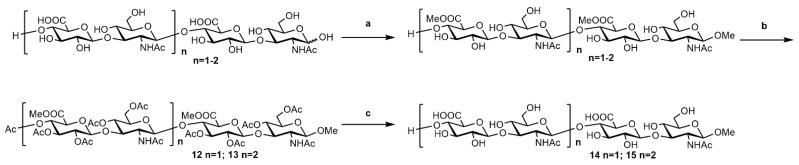
Synthesis of o-HA_4_OMe (**14**) and o-HA_6_OMe (**15**). Reagents and conditions: (**a**) 0.5 M HCl, MeOH, 0 °C to rt, overnight; (**b**) Ac_2_O, Et_3_N, DMAP, DCM, 2 h, 21% for **12**, 19% for **13**. (**c**) LiOH/H_2_O_2_, THF/H_2_O, −5 °C to rt, then 4 M NaOH, MeOH, 0 °C to rt, 85% for **14**, 82% for **15**.

**Figure 5 molecules-27-05619-f005:**
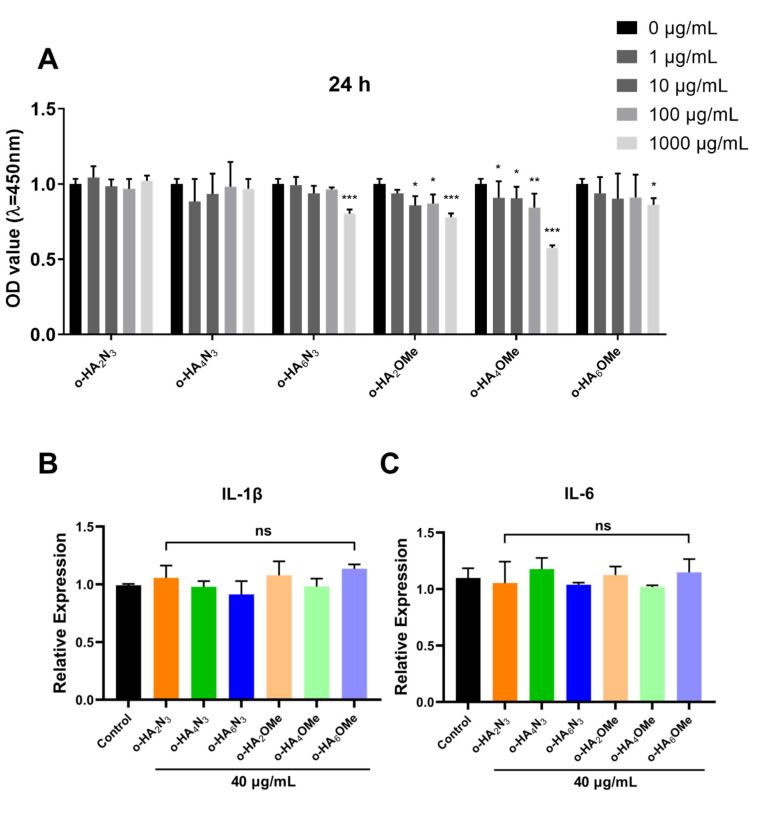
Evaluation of the cytotoxicity and pro-inflammation of o-HA derivatives in ATDC5 cells. (**A**) ATDC5 cells were seeded into 96-well plates. After 12 h of incubation, cells were treated with different concentrations (0, 1, 10, 100, 1000 μg/mL) of o-HA derivatives for 24 h. Next, cell viability was examined by the CCK-8 assay. (**B**,**C**) RT-qPCR was used to confirm the expression levels of pro-inflammatory cytokines (IL-1β, IL-6) after co-culture with ATDC5 cell of o-HA derivatives (40 μg/mL) for 24h. * *p* < 0.05, ** *p* < 0.01, *** *p* < 0.001 vs. each 0 μg/mL group; ns = not statistically significant vs Control group.

**Figure 6 molecules-27-05619-f006:**
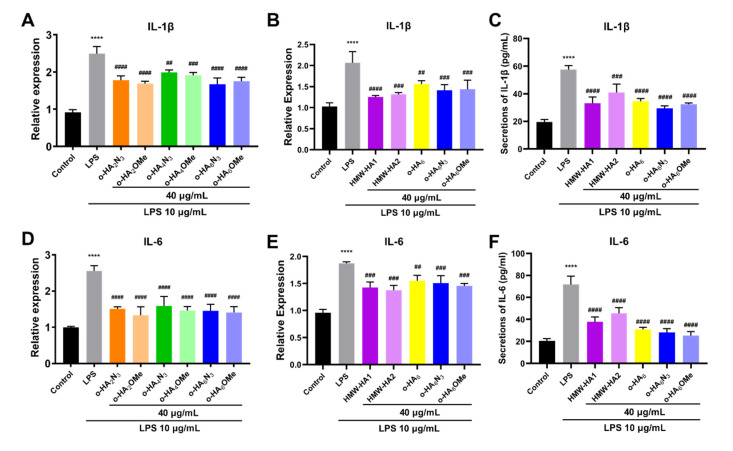
Effects of different HA derivatives on LPS-induced inflammatory responses in ATDC5 cells. ATDC5 cells in the LPS group were stimulated with LPS (10 μg/mL) for 24 h and then cultured in a medium containing 10% FBS for another 24 h. For different HA derivatives groups, ATDC5 cells were treated with LPS (10 μg/mL) for 24 h followed by treatment with corresponding samples (40 μg/mL) for an additional 24 h, respectively. (**A**–**C**) IL-1β mRNA expression levels and cytokine level were measured by RT-qPCR and ELISA. (**D**–**F**) IL-6 mRNA expression levels and cytokine level were measured by RT-qPCR and ELISA. **** *p* < 0.0001 vs. Control group; ^#^
*p* < 0.05, ^##^
*p* < 0.01, ^###^ *p* < 0.001, ^####^
*p* < 0.0001 vs. LPS group.

**Figure 7 molecules-27-05619-f007:**
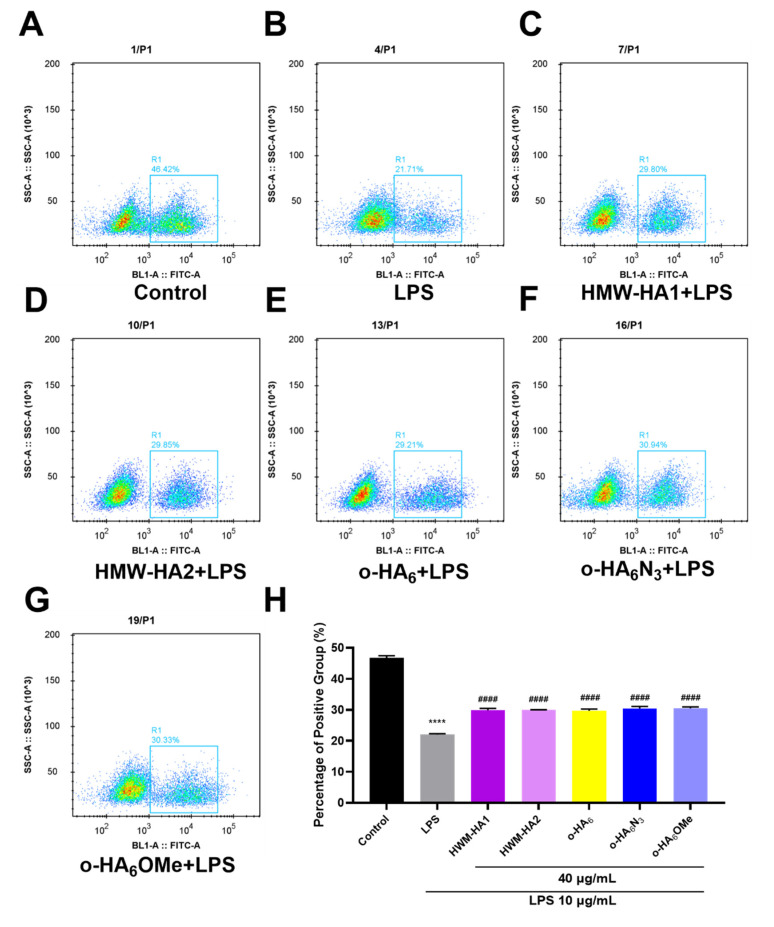
Effects of different HA derivatives on cell proliferation after inflammatory injury. (**A**–**H**) ATDC5 cells in the LPS group were stimulated with LPS (10 μg/mL) for 24 h and then cultured in the medium containing 10% FBS for another 24 h. For different HA derivatives groups, ATDC5 cells were treated with LPS (10 μg/mL) for 24 h and different HA derivatives (40 μg/mL) for an additional 24 h, respectively. (**A**–**G**) The positive cell ratio was measured by EdU flow cytometry assay. (**H**) Histogram quantification was performed for the triplicate results. **** *p* < 0.0001 vs. Control group; ^####^
*p* < 0.0001 vs. LPS group.

**Figure 8 molecules-27-05619-f008:**
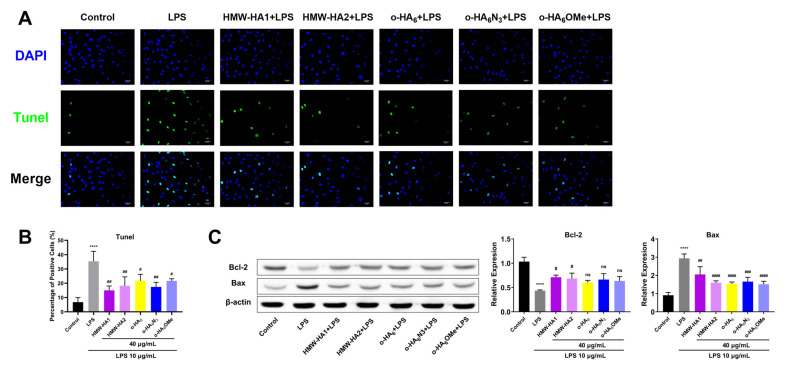
Anti-apoptotic effects of different HA derivatives on ATDC5 inflammatory injury. ATDC5 cells in the LPS group were stimulated with LPS (10 μg/mL) for 24 h and then cultured in a medium containing 10% FBS for another 24 h. For different HA derivatives groups, ATDC5 cells were treated with LPS (10 μg/mL) for 24 h and different HA derivatives (40 μg/mL) for an additional 24 h, respectively. (**A**) Cell apoptotic pattern was examined by TUNEL assay. (**B**) Positive cell counts were plotted for each treatment group for three independent replicates. (**C**) Western blot assay was used to detect the protein expression levels of Bcl-2 and Bax after treatment with different HA derivatives. **** *p* < 0.0001 vs. Control group; ^#^
*p* < 0.05, ^##^
*p* < 0.01, ^###^
*p* < 0.001, ^####^
*p* < 0.0001, ns = not statistically significant vs. LPS group.

**Figure 9 molecules-27-05619-f009:**
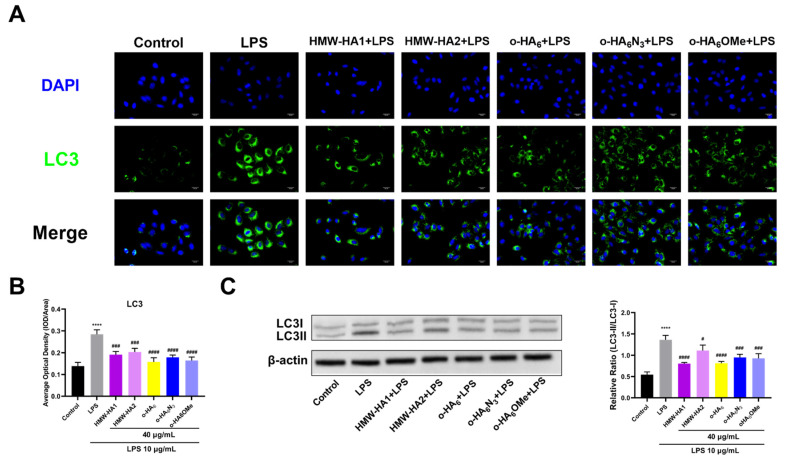
Anti-autophagy effect of different HA derivatives on ATDC5 cells after inflammatory injury. ATDC5 cells in the LPS group were stimulated with LPS (10 μg/mL) for 24 h and then cultured in the medium containing 10% FBS for another 24 h. For different HA derivatives groups, ATDC5 cells were treated with LPS (10 μg/mL) for 24 h and different HA derivatives (40 μg/mL) for an additional 24 h, respectively. (**A**) Cell autophagy pattern was assessed by LC3 immunofluorescence analysis. (**B**) Relative fluorescence intensity was plotted for each treatment group for three independent replicates. (**C**) LC3 protein expression level and LC3-II/LC3-I ratio were detected by Western Blot assay. **** *p* < 0.0001 vs. Control group; ^#^
*p* < 0.05, ^###^
*p* < 0.001, ^####^
*p* < 0.0001 vs. LPS group.

**Figure 10 molecules-27-05619-f010:**
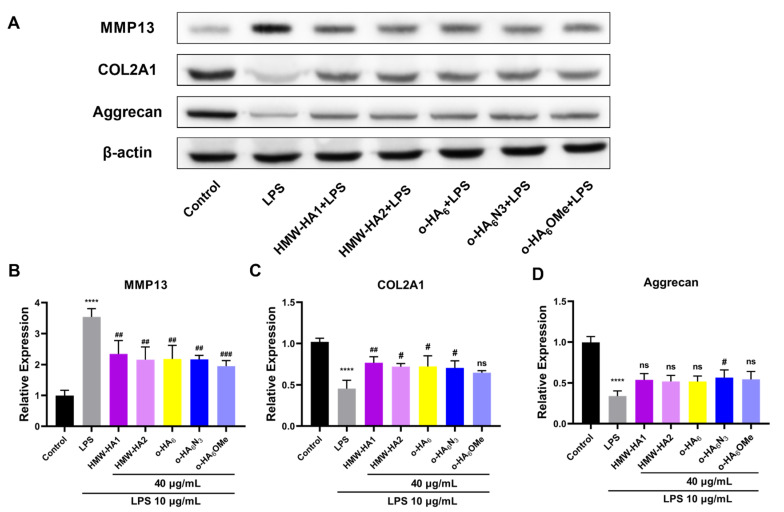
Effects of different HA derivatives on the degradation of ECM caused by ATDC5 inflammatory injury. ATDC5 cells in the LPS group were stimulated with LPS (10 μg/mL) for 24 h and then cultured in the medium containing 10% FBS for another 24 h. For different HA derivatives groups, ATDC5 cells were treated with LPS (10 μg/mL) for 24 h and different HA derivatives (40 μg/mL) for an additional 24 h, respectively. (**A**–**D**) The protein expression levels of ECM degradation related indicators (MMP13, Aggrecan and COL2A1) were detected by Western blot assay.. **** *p* < 0.0001 vs. Control group; ^#^
*p* < 0.05, ^##^
*p* < 0.01, ^###^
*p* < 0.001, ns = not statistically significant vs. LPS group.

## Data Availability

The data presented in this study are available on request from the corresponding author. The data are not publicly available due to data protection protocol among researchers.

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
