# Peer review of "Hyaluronic Acid Oligosaccharide Derivatives Alleviate Lipopolysaccharide-Induced Inflammation in ATDC5 Cells by Multiple Mechanisms"

_molecules, 2022, doi:10.3390/molecules27175619_

Round 1

Reviewer 1 Report

The manuscript entitled "Hyaluronic Acid Oligosaccharide Derivatives Alleviate Lipopolysaccharide-induced Inflammation in ATDC5 Cells by Multiple Mechanisms" by Huang et al. outlines the modulatory potential of the newly synthesized series of oligosaccharides of HA (o-HAs) on a chondrocyte inflammatory model established by the lipopolysaccharide (LPS)-challenged ATDC5 cells. The study is well performed and organized, obtained results indicate the potential of oligosaccharides of HA application in osteoarthritis treatment. Overall, I suggest accepting the proposed paper in the form as it is.

English language and style are fine/minor spell check and typos are required.

Reviewer 2 Report

molecules-1761891

Hyaluronic Acid Oligosaccharide Derivatives Alleviate Lipopolysaccharide-induced Inflammation in ATDC5 Cells by Multiple Mechanisms

Hesuyuan Huang , Xuyang Ding , Dan Xing , Jianjing Lin , Zhongtang Li , Jianhao Lin

The work is well design and solid. It can be accepted after revision.

Some points are

The novelty of the work should be better addressed.

The size of the figures should be increased. For example it is not necessary to write the reactions in one line. You can increase the size and the products (after 4-5 steps) can be written in a different line

Figure 5 should be better discussed, especially about the differences. Also, why the error bars are so different? Please explain

Please comment about the processes used for cleaning each product from byproducts or reactants. The yield is high but about 85%.

The only comment about higher quantities is that the process is with few steps. Although it is desirable please comment about the increasing quantities also.

Reviewer 3 Report

In this study, the authors have produced a number of hyaluronic acid oligosaccharide derivatives, and have examined their effects on OA progression in an LPS-challenged chondrocyte cell line in comparison to an untreated one. The authors conclude that the synthesized derivatives reduce the inflammatory response. The findings in this study challenge previous findings on the roles of HMW HA and fragmented HA in OA progression. The authors should emphasize the novelty of the current work in view of previous studies. The paper may be acceptable for publication after major structural changes.

1.           The study doesn´t provide any analytical methods to determine the biocompatibility of the derivatives and how their presence affects the supramolecular structure of the extracellular matrix of joint tissues. At a minimum, this should be discussed.

2.           The influence of derivatives in the inflammatory response in comparison to HMW HA has to be determined.

3.           To establish a physiologically relevant in vitro model of OA-related inflammation, the authors should utilize a combination of PAMPs (LPS or Listeria bacteria) and a DAMP (such as hyaluronan (HA) fragments) that would induce IL-1b production. How do the authors explain the IL-1b reducing effect of the derivates?

4.           Most studies are carried out using only one cell line. More cell lines are needed to confirm functional and mechanical studies.

5.           HA fragments are generated by reactive oxygen species associated with inflammation and enzymatically, by hyaluronidases, during OA progression. The authors use bovine testis hyaluronidase which has been shown to be contaminated with large amounts of bFGF (Rahmanian M. and Heldin P, Int. J. Cancer:97, 601-607, 2002). The authors have to investigate that the effects of the derivatives are not due to contamination by bFGF or other growth factors or cytokines.

Round 2

Reviewer 2 Report

The authors have well addressed my comments. The manuscript can be accepted.

The authors should change the style in References.

Reviewer 3 Report

My concerns were answered, and the revised version is in a better format.